# Sex-Specific Differences in Fat Storage, Development of Non-Alcoholic Fatty Liver Disease and Brain Structure in Juvenile HFD-Induced Obese Ldlr-/-.Leiden Mice

**DOI:** 10.3390/nu11081861

**Published:** 2019-08-10

**Authors:** Sophie A.H. Jacobs, Eveline Gart, Debby Vreeken, Bart A.A. Franx, Lotte Wekking, Vivienne G.M. Verweij, Nicole Worms, Marieke H. Schoemaker, Gabriele Gross, Martine C. Morrison, Robert Kleemann, Ilse A.C. Arnoldussen, Amanda J. Kiliaan

**Affiliations:** 1Department of Anatomy, Radboud university medical center, Donders Institute for Brain, Cognition and Behaviour, Preclinical Imaging Center PRIME, 6500 HB Nijmegen, The Netherlands; 2Department of Metabolic Health Research, Netherlands Organisation for Applied Scientific Research (TNO), 2333 CK Leiden, The Netherlands; 3Human and Animal Physiology, Wageningen University, 6700 AH Wageningen, The Netherlands; 4Reckitt Benckiser Health, Mead Johnson Pediatric Nutrition Institute, 6545 CJ Nijmegen, The Netherlands; 5Department of Vascular Surgery, Leiden University Medical Center, 2300 RC Leiden, The Netherlands

**Keywords:** obesity, juvenile, sex

## Abstract

Background: Sex-specific differences play a role in metabolism, fat storage in adipose tissue, and brain structure. At juvenile age, brain function is susceptible to the effects of obesity; little is known about sex-specific differences in juvenile obesity. Therefore, this study examined sex-specific differences in adipose tissue and liver of high-fat diet (HFD)-induced obese mice, and putative alterations between male and female mice in brain structure in relation to behavioral changes during the development of juvenile obesity. Methods: In six-week-old male and female Ldlr-/-.Leiden mice (n = 48), the impact of 18 weeks of HFD-feeding was examined. Fat distribution, liver pathology and brain structure and function were analyzed imunohisto- and biochemically, in cognitive tasks and with MRI. Results: HFD-fed female mice were characterized by an increased perigonadal fat mass, pronounced macrovesicular hepatic steatosis and liver inflammation. Male mice on HFD displayed an increased mesenteric fat mass, pronounced adipose tissue inflammation and microvesicular hepatic steatosis. Only male HFD-fed mice showed decreased cerebral blood flow and reduced white matter integrity. Conclusions: At young age, male mice are more susceptible to the detrimental effects of HFD than female mice. This study emphasizes the importance of sex-specific differences in obesity, liver pathology, and brain function.

## 1. Introduction

Men and women differ in regard to the storage and distribution of adipose tissue [1,2], but studies examining adipose tissue related sex differences are limited [3,4,5,6,7,8,9,10,11]. The prevalence of obesity in adults almost tripled since 1975, with an overall even higher prevalence in women (15%) compared to men (11%) [12]. Men tend to accumulate more visceral adipose tissue (VAT), while women develop more subcutaneous adipose tissue (SAT) [13]. Excessive storage of VAT is thought to contribute to metabolic abnormalities, a release of inflammatory mediators and cardiovascular risk factors, whereas fat storage in SAT is considered to be safer [14,15,16]. Differences in the VAT/SAT-ratio have been reported between men and women, and these differences disappear as the amount of VAT increases after menopause [17]. 

With respect to sex-specific differences in the context of diet-induced obesity, Hwang and colleagues reported that male mice are more sensitive to high fat diet (HFD) feeding when it comes to weight gain and metabolic alterations, but also regarding learning and hippocampal synaptic plasticity [11]. This suggests that fat handling and storage across adipose tissue depots may differ among sexes, and points to a role of sex in obesity-associated brain function and brain development [18,19,20]. 

The rise of obesity in childhood and adolescence is of particular concern as it appears to affect the developing brain. For instance, a study in 6 to 24 months old infants observed an impairment in cognitive abilities in obese infants compared to their lean peers [21]. This early life effect of obesity on cognition has been described in numerous rodent studies [3,4,5,11]. Thus, at juvenile age, brain structure and function of subjects appear to be particularly susceptible to the effects of obesity [7,8]. However, much remains unknown about this period of sensitivity, and potential dependent effects on cognition.

In general, little is known about putative sex-dependent differences in metabolic parameters and their relation to brain structure and function in juvenile obesity. Therefore, this study investigates sex-specific variations during HFD-induced obesity in metabolically active tissues (liver and adipose tissue), as well as potential differences in brain structure and function, i.e., the relationship to behavioral changes during the development of juvenile obesity. To examine this, young (6-week old) low density lipoprotein receptor-deficient (Ldlr-/-) Leiden (Ldlr-/-.Leiden) mice were fed either an energy-dense obesogenic HFD or low-fat reference AIN93G diet for 18 weeks [22,23,24,25,26]. When fed a HFD, adult Ldlr-/-.Leiden mice are prone to develop diet-induced obesity, adipose tissue inflammation, and insulin resistance. Adult Ldlr-/-.Leiden mice subsequently develop non-alcoholic fatty liver disease (NAFLD) with human-like features of non-alcoholic steatohepatitis (NASH) [25,27]. In addition, adult mice can develop obesity-associated cognitive dysfunction and changes in functional connectivity in the brain [28]. This study examines sex-specific differences in metabolically active tissues and cognitive function in young mice along the course of juvenile diet-induced obesity. Thereby, our study extends knowledge on specific sex-differences in context of juvenile obesity, which may eventually help to develop sex-specific interventions.

## 2. Materials and Methods 

### 2.1. Animals, Diets and Study Design

Both female and male Ldlr-/-.Leiden mice, originating from a specific-pathogen-free (SPF) breeding stock at TNO (TNO-Metabolic Health Research, Leiden, the Netherlands), were used in this study. Mice were group- housed in digitally ventilated cages (DVC, Tecniplast S.p.A., Buguggiate, Italy) in conventional animal rooms (relative humidity 50–60%, temperature ~21°C, light cycle 7 a.m.–7 p.m.) in the preclinical imaging center (PRIME) at the central animal laboratory, Radboud university medical center, The Netherlands. The mice had *ad libitum* access to food and acidified tap water. The experimental approach was approved by an independent institutional ethical committee on animal care and experimentation (Zeist, the Netherlands), approval number DEC3682. We minimized the number of animals used in the experiments as statistical power analyses were conducted, and additionally the ARRIVE guidelines of experimental performance and reporting were met [29].

At weaning (3 weeks old), 24 female and 24 male Ldlr-/-.Leiden pups were randomly divided over cages (4 animals per cage). Post-weaning, all mice were fed a semi-synthetic low-fat reference diet, AIN93G for three weeks (D10012G, Research Diets, New Brunswick, USA, diets presented in Table 1). At 6 weeks of age, the mice were randomly divided into two dietary groups. The first group was kept on the AIN93G diet throughout the whole experiment, as control condition (AIN93G, N = 24, 12 mice per sex). The second group, representing juvenile obesity, received a HFD from 6 weeks of age for 18 weeks (HFD18 (45% kcal fat); D12451, Research Diets; N = 24, 12 mice per sex). 

Diet-induced changes in blood plasma, systolic blood pressure, cognition, brain structure and function were measured in both female and male Ldlr-/-.Leiden mice according to the experimental scheme presented in Figure 1.

### 2.2. Plasma Analyses

After 5 h of fasting (8 a.m.–1 p.m.), blood was collected by a tail vein incision, after which EDTA plasma was isolated. Blood glucose levels were measured at the time of blood sampling using a hand-held glucometer (FreeStyle Freedom Lite; Abbott Diabetes Care, Hoofddorp, the Netherlands). Plasma analyses were performed using standardized protocols and assays [25,30,31]. Total plasma levels of cholesterol and triglycerides were determined using enzymatic assays (CHOD-PAP and GPO-PAP respectively; Roche Diagnostics, Almere, the Netherlands). Plasma insulin, leptin, adiponectin, liver-derived insulin-like growth factor I (IGF-I), insulin-like growth factor binding protein (IGFBP) and serum amyloid A (SAA) were measured using standardized ELISA kits (Mercodia, Uppsala, Sweden for insulin; R&D Systems, Inc., Minneapolis, MN, USA for leptin, adiponectin, IGF-I and IGFBP (Life Technologies, Bleiswijk, the Netherlands for SAA).

### 2.3. Sacrifice

Anaesthetized mice (3.5% isoflurane, Nicholas Primal (I) limited, London, UK) were sacrificed by transcardial perfusion with 0.1 M phosphate-buffered saline (PBS, 7.3 pH, room temperature). Brain, liver and three fat depots (mesenteric, perigonadal and inguinal) were immediately harvested. Subsequently, these three different fat depots and the liver were weighed. Additionally, we determined percentage of body fat (total fat depots weight divided by body weight (24 weeks old)), and the VAT/SAT-ratio (mesenteric weight divided by inguinal weight). For immunohistochemical purposes, the right brain hemisphere, three fat depots and the liver were stored in 0.1 M PBS with 0.01% sodium azide at 4 °C, after being postfixed in 4% paraformaldehyde (7.2 pH, 4 °C) for 24 h. 

### 2.4. Liver Analyses

The liver was cut into 3 µm-thick cross sections and stained with haematoxylin and eosin for histopathological analysis of NAFLD. Development of NAFLD was scored in two cross-sections per mouse, by a board-certified pathologist (blinded for condition) using a well-established scoring system for rodent NAFLD [32] as previously described [33]. Briefly, macrovesicular steatosis, microvesicular steatosis and hepatocellular hypertrophy were expressed as the percentage of the total liver section affected. Hepatic inflammation was quantified by counting the number of inflammatory cell clusters in 5 non-overlapping fields per specimen at 100x magnification (view size 2.3 mm^2^) and expressed as the number of inflammatory aggregates per field.

### 2.5. Analyses in White Adipose Tissue

Perigonadal, mesenteric and inguinal adipose tissue were fixed in formalin, embedded in paraffin and sliced in 5 μm sections. Sections were stained with haematoxylin phloxine saffron and digitised using a slide scanner (Aperio AT2, Leica Biosystems, Amsterdam, The Netherlands). Adipose tissue morphometry (average adipocyte size and adipocyte size distribution) was analyzed using Adiposoft [34] an open-source automated plug-in for the image processing package Fiji [35] for ImageJ [36]. Adipose tissue inflammation was analyzed by counting the number of crown-like structures (CLS; the histological hallmark of adipose tissue inflammation [37]), in the same fields used for the morphometry analysis and expressed as number of CLS per 1000 adipocytes. 

### 2.6. Behavioral and Cognitive Tests

#### 2.6.1. Digitally Ventilated Cages

To assess general activity during day and night, digitally ventilated cages (DVC, Tecniplast S.p.A., Buguggiate, Italy) were used [38]. These cages monitor home-cage activity (at cage level) 24 h per day, and seven days per week. Day and night activity was indicated by activation of the 12 cage-sensors (arbitrary units (a.u.)) using DVC data obtained during weekends.

#### 2.6.2. Systolic Blood Pressure Measurements 

To provide more information about arterial functioning, systolic blood pressure (SBP) was measured using a computerized and warmed tail-cuff plethysmography device (IITC Life Scientific Instruments, Woodland Hill, CA, USA). Habituation to the procedure occurred one day prior to the actual measurement days. In the following 2 days, 6 measurements were conducted, of which the average was used for further statistical analysis. Two male mice fed a HFD were excluded from analyses due to stress during the blood pressure measurements.

#### 2.6.3. Open Field

The open field (OF) test was assessed to measure locomotor activity and explorative behavior, following the protocol previously described [39,40,41]. The mice were individually placed in a square transparent plastic box (45 × 45 × 45 cm) facing north. Activity in the OF was automatically monitored by EthoVision XT10.1 (Noldus, Wageningen, the Netherlands), and the total duration of walking, sitting, wall leaning, rearing and grooming behavior was manually scored. 

#### 2.6.4. Morris Water Maze

The Morris water maze (MWM) was conducted to assess spatial learning and memory (hippocampal-dependent task) according to a previously described protocol [41]. During the acquisition phase, a platform (Ø = 8 cm) was placed in a circular pool (Ø = 108 cm) one centimeter below the water surface in the south-western quadrant. The water was made opaque by adding milk powder (water temperature = 21–22 °C). On each wall, a spatial cue was located at 0.5 m distance from the pool. During 4 consecutive days, the mice were placed in the pool at 4 different locations (south, north, east, west). Using the four different visual cues, mice navigated faster to the platform with every trial (max swimming time 120 s; 30 s on platform; inter-trial interval 60 min). All trials were recorded using EthoVision XT10.1 (Noldus) and latency to platform was scored as a measure for spatial learning. At the end of the fourth day, a probe trial was performed in which the platform was removed. Time spent in the quadrant where the platform used to be, and the number of platform crossings were used as a measure of spatial memory. 

#### 2.6.5. MRI Experiments

Magnetic resonance imaging (MRI) measurements were performed with a 11.7 Tesla Biospec Avance III small animal MR system (Bruker BioSpin, Ettlingen, Germany) equipped with an actively shielded gradient set of 600 mT/m and operating with a Paravision 6.1 software platform. A circular polarized volume resonator was used for signal transmission. Furthermore, an actively decoupled mouse brain quadrature surface coil with integrated combiner and preamplifier was used for signal receiver (Bruker BioSpin). All MRI experiments were executed according to previously described protocols by Zerbi and colleagues [42,43]. Mice were anaesthetized with isoflurane (Nicholas Primal (I) limited, London, UK; 3.5% for induction for 2 min; approximately 1.8% during the actual scan in a mixture of oxygen and medical air (1:2)). To minimize movement artifacts, mice were placed in a stereotactic device to immobilize their heads. Respiration, monitored with pneumatic cushion respiratory monitoring system (Small Animal Instruments Inc., Stony Brook, NY, USA) was kept constant. Moreover, body temperature remained stable at 37 °C by using an airflow device and a rectal thermometer. 

Standard adjustments and a gradient echo in axial, sagittal and coronal directions were made, providing an anatomical reference of the mouse brain. To obtain information about cerebral blood flow (CBF), arterial spin labeling (ASL) was performed. Furthermore, functional connectivity (FC) was assessed using resting state fMRI (rsfMRI), and diffusion tensor imaging (DTI) was performed to indicate gray matter (GM) and white matter (WM) integrity [42]. Imaging parameters are represented in Table 2.

#### 2.6.6. Immunohistochemistry 

According to a previously described standardized protocol for free-floating labeling immunohistochemistry [41], 30 µm-thick coronal brain sections of the fixed right brain hemisphere were made, using a sliding microtome, equipped with an object table for freezing sectioning at −60 °C (Microm HM 440, Walldorf, Germany). In short, we examined: (a) the amount of glucose transporter 1 (GLUT-1) as a measure of blood vessel integrity, (b) doublecortin (DCX) expressed in immature neurons as a measure of neurogenesis, and (c) reactive microglia by ionized calcium binding adapter molecule-1 (IBA-1) to examine the level of neuroinflammation. Therefore, 30 µm-thick sections were incubated overnight at room temperature with primary antibody either GLUT-1 (rabbit-anti-GLUT-1 (1:40,000), Chemicon International, Inc., Temecula, CA, USA), DCX (goat-anti-DCX (1:4000), Santa Cruz Biotechnology Inc., Dallas, USA), or IBA-1 (goat-anti-IBA-1 (1:2000), Abcam, Cambridge, UK). Thereafter, the sections were incubated for 90 min with the secondary antibody, donkey-anti-goat (1:1500 Jackson Immuno Research Europe Ltd., Suffolk, UK), or donkey-anti-rabbit (1:1500 Jackson), and slices were pre-incubated with DAB-Ni [44]. Finally, three sections per mouse per tissue (cortex, thalamus and hippocampus) were used for quantification. DCX positively stained cells were manually counted by 2 blinded raters at 40x magnification and the average was used for further statistical analysis. The three sections stained for GLUT-1 and IBA-1 were photographed at 5x magnification with an Axio Imager A2 microscope (Zeiss, Köln, Germany). An automatic and standardized ImageJ (JAVA-based public domain, National Institutes of Health, Bethesda, MD, USA) protocol was used to quantify number of positive glucose transporters in blood vessels or reactive microglia per mm^2^.

### 2.7. Statistical Analysis

Data was analyzed with SPSS25 (IBM SPSS Statistics 25, IBM Corporation, Armonk, New York, USA). All results are expressed as mean ± standard error of mean (SEM). Means were analyzed using repeated measures or multivariate analysis of variance (ANOVAs) with Bonferroni correction for multiple testing. Correlation tests were performed using Pearson correlations. Diet and sex-specific effects were referred to as significant if the *p*-value was lower than 0.050 (*p*- and *F*-values are presented in Table 3 and Table 4).

## 3. Results

### 3.1. Sex-Specific Differences in Metabolically Active Tissues

#### 3.1.1. Male Mice Were Significantly Heavier than Female Mice, but no Differences Were Observed in HFD-Induced Body Weight Gain

The body weight of female mice fed the reference AIN93G diet was significantly lower than that of male mice. HFD feeding significantly increased the body weight of both female and male mice compared to their AIN93G diet fed controls. Male mice fed HFD were significantly heavier than female mice on HFD (Figure 2A). However, body weight gain on HFD, which was corrected for the regular growth-related weight gain on the reference AIN93G diet, was not significantly different between sexes in the HFD dietary groups (Figure 2B). Furthermore, no significant differences in kilo caloric (kcal) intake (at cage level) were found for either diet or sex-specific effect, indicating a comparable energy intake for all groups (Figure 2C,D). The percentage of body fat was comparable between female and male mice fed the AIN93G diet. HFD feeding significantly increased the percentage of body fat only in female mice (Figure 2E). In addition, the VAT/SAT-ratio was comparable between sexes and diets (Figure 2F). 

#### 3.1.2. Female Mice Had Lower Glucose and Adiponectin Levels, and upon HFD Feeding Lower Insulin, Cholesterol and Triglyceride Levels than Male Mice 

We next analyzed circulating levels of fasting insulin, glucose, triglycerides, cholesterol, leptin and adiponectin. Female mice had lower fasting plasma insulin levels than male mice on AIN93G diet. In female mice, insulin levels hardly increased upon HFD feeding, whereas males showed pronounced hyperinsulinemia (Figure 3A). However, blood glucose levels did not change upon HFD in female and male mice, and glucose levels of female mice remained lower than in male mice upon HFD feeding (Figure 3B). Plasma triglycerides were comparable between female and male mice on AIN93G. Only in male mice HFD feeding induced higher plasma triglyceride levels, and were significantly higher than in female mice fed HFD (Figure 3C). Plasma cholesterol levels on AIN93G diet were lower in female mice compared to male mice. HFD feeding increased plasma cholesterol levels in both female and male mice, but in male mice to a significantly higher level than in female mice (Figure 3D). Both sexes had comparable plasma leptin levels on AIN93G. HFD treatment increased leptin levels comparably in female and male mice (Figure 3E). Lastly, female mice on an AIN93G diet had much higher adiponectin levels than male mice on the same diet. HFD feeding did not significantly alter plasma adiponectin levels in female and male mice (Figure 3F).

#### 3.1.3. Female Mice on HFD Show a Predominant Fat Storage within the Perigonadal Fat Depot, and Exert a Lower Degree of Adipose Tissue Inflammation 

Mesenteric fat mass was similar in male and female mice fed AIN93G. Representative photomicrographs of mesenteric adipose tissue are provided in Figure 4A–D. HFD feeding led to a significantly increase in mesenteric fat weight in male mice only (Figure 4E). Within the mesenteric depot, sex-specific differences were observed regarding distribution of adipocyte sizes. On both AIN93G diet and HFD, female mice had a higher percentage of smaller adipocytes and a lower percentage of larger adipocytes compared with male mice (Figure 4G,H). HFD feeding shifted the distribution of adipocyte size towards larger adipocytes in both female and male mice compared to AIN93G fed controls (Figure 4I,J). Lastly, we analyzed the number of crown-like structures (CLS) as a measure of adipose tissue inflammation. Female mice fed an AIN93G diet had a lower number of mesenteric CLS than male mice, and a significant increase in CLS in the mesenteric fat depot upon HFD feeding was observed in male mice only (Figure 4F). 

Perigonadal fat weight was significantly lower in female than in male mice on AIN93G. HFD treatment strongly increased the weight of this fat depot in female mice relative to AIN93G-fed controls. In male mice, no significant increase in perigonadal fat weight was observed upon HFD feeding and the mass of this depot was significantly lower than in HFD-fed female mice (Figure 5A). Moreover, female mice had a significantly higher percentage of larger adipocytes and a lower percentage of smaller adipocytes than male mice, either fed a AIN93G diet or HFD (Figure 5C,D). In female mice, HFD feeding lowered the percentage of small adipocytes (2000–4000 µm^2^, Figure 5E), while adipocyte size distribution was not affected by HFD feeding in male mice (Figure 5F). The number of CLS within the perigonadal fat depots was significantly higher in male than female mice on both AIN93G diet and HFD. However, HFD feeding did not significantly affect the number of CLS in female or male mice relative to their respective AIN93G controls (Figure 5B).

No differences in inguinal fat mass were observed between male and female mice on AIN93G. HFD feeding increased the inguinal fat mass in both female and male mice with no difference in mass between sexes (Figure 5G). Similarly, there were no differences between sexes in adipocyte size distribution. HFD-feeding induced a shift towards a lower percentage of smaller adipocytes, and an increased percentage of larger adipocytes in both female and male mice (Figure 5I,J). The number of CLS within the inguinal fat depots did not significantly differ between male and female mice fed AIN93G diet. HFD feeding significantly increased the number of CLS in the inguinal depot in male mice only (Figure 5H). Altogether, the storage of fat across the white adipose tissue (WAT) depots differed among male and female mice, and only male mice developed adipose tissue inflammation in response to HFD feeding.

#### 3.1.4. Increased Macrovesicular Steatosis and Hepatic Inflammation in HFD Fed Female Mice

We analyzed liver tissue to examine sex differences in the development of NAFLD, and representative images of liver tissue are presented in Figure 6A–D. The liver weights of female and male mice fed an AIN93G diet were comparable. HFD feeding increased liver weight in only male mice suggesting that male mice developed steatosis (Figure 6E). Histopathological analysis of the livers revealed that microvesicular steatosis was low (about 5–10% of cross-sectional liver area) on AIN93G diet, and did not differ significantly between female and male mice. While HFD feeding did not induce microvesicular steatosis in female mice, male mice showed pronounced development of microvesicular liver steatosis (about 30%) (Figure 6F). Counterintuitively, high levels of macrovesicular steatosis were observed in AIN93G treated female mice (about 50%), which was significantly higher than in male mice (about 20%) on AIN93G. Surprisingly, HFD feeding did not significantly affect macrovesicular steatosis in either sex, and female mice even showed an insignificant reduction (Figure 6G). In addition, hepatocellular hypertrophy on AIN93G was higher in female mice than in male mice, but was not further increased by HFD in female mice. By contrast, HFD feeding in male mice strongly increased hepatocellular hypertrophy. Consistent with the observed development of HFD-induced liver steatosis, male mice showed significantly higher levels of hepatocellular hypertrophy than in female mice fed a HFD (Figure 6H). Lastly, female mice showed high levels of hepatic inflammation on the reference AIN93G diet, whereas young male mice fed AIN93G diet showed relatively low levels. Hepatic inflammation in female and male mice was not significantly increased by HFD feeding (Figure 6I). 

In summary, young female mice were characterized by (a) predominant fat storage within the perigonadal and inguinal fat depots, (b) neglectable levels of adipose tissue inflammation levels on AIN93G that remain low even when fed a HFD, (c) presence of macrovesicular steatosis in the liver that is not enhanced by HFD, and (d) high liver inflammation levels on AIN93G that were not further increased by HFD feeding. In young male mice, HFD induced (a) fat storage within the mesenteric and inguinal depots, (b) pronounced adipose tissue inflammation in several adipose tissue depots, and (c) microvesicular steatosis in the liver. There were thus remarkable differences between young female and young male mice in fat storage and inflammation levels in both the adipose tissue and the liver on the low-fat AIN93G reference diet, but also upon HFD feeding, and it should be emphasized that young Ldlr-/-.Leiden mice (i.e., in context of juvenile obesity) responded differently to HFD than adult mice in reported studies [25,28,31]. 

### 3.2. Sex-Specific Alterations and HFD-Indued Juvenile Obesity in Brain Structure, Function and Behavior

We examined whether sex differences exist in SBP, activity and explorative behavior in the open field test, or spatial learning and memory abilities obtained in the MWM on reference AIN93G diet or in response to 18 weeks of HFD feeding. No significant effects were found concerning SBP or open field parameters like locomotor activity, velocity and explorative behavior (Appendix A). In addition, we found no significant differences in spatial learning and memory abilities obtained in the MWM (Appendix A). In DVC, in which day and night activity are automatically monitored, we found no significant differences in day and night activity between male and female mice fed a reference AIN93G diet (Figure 7A,B). However, we observed that female mice fed a HFD were slightly less active during daytime compared to male mice on the same diet (Figure 7A). 

#### Male Mice Fed a HFD Have a Lower Cerebral Blood Flow and Decreased White Matter Integrity

We aimed to elucidate the role of the cerebrovasculature in relation to adiposity and brain function. Therefore, we examined blood vessel integrity via GLUT-1 immunohistochemistry, and we determined CBF levels of these young mice. No significant differences were found for either sexes or diets in GLUT-1 analyses (Appendix A). The CBF levels were comparable between female and male mice on a reference AIN93G diet. Of note, CBF was significantly reduced in the hippocampus in only male mice upon HFD feeding indicating a sex-specific effect (Figure 7C).

No significant differences were found in neurogenesis (DCX; data not shown), neuroinflammation (IBA-1, Appendix A), nor functional connectivity for either diet or sex. DTI was obtained to indicate gray matter (Figure 7D–H) and white matter integrity (Figure 7I–K) in several brain regions. In detail, higher mean diffusivity (MD) levels are associated with decreased gray matter integrity. Female mice showed significantly lower MD levels in the auditory cortex (Figure 7D), somatosensory cortex (Figure 7E), visual cortex (Figure 7F), and motor cortex (Figure 7H) compared to male mice, both fed the same reference AIN93G diet. No alterations in MD levels were found upon HFD feeding for female and male mice compared to the reference diet (Figure 7E–G). Female mice on HFD showed a lower MD in the auditory cortex compared to male mice on HFD (Figure 7D). Furthermore, lower fractional anisotropy (FA) levels might indicate decreased white matter integrity. Female mice on AIN93G diet had a lower FA level within the fornix compared to male mice fed the same diet (Figure 7I). HFD feeding significantly decreased the FA level in the motor cortex in specifically male mice (Figure 7K). In addition, in the hippocampus (Figure 7J) and motor cortex (Figure 7K) of female mice on HFD, the FA levels were significantly higher compared to male mice on HFD. 

Summarizing, young male mice showed decreased gray matter integrity (auditory cortex, somatosensory cortex, visual cortex and motor cortex) and white matter integrity (motor cortex and hippocampus) compared to young female mice both on the same reference AIN93G diet. Eighteen weeks of HFD feeding slightly decreased white matter integrity in the motor cortex in young male mice. Thereby, these results imply that young male mice seem to be slightly more prone to HFD-induced structural brain changes than young female mice, indicated by decreased CBF and white matter integrity. 

## 4. Discussion

In this study, we examined sex-specific differences during the development of juvenile obesity in HFD-fed Lldr-/-.Leiden mice with particular focus on metabolic risk factors and organs as well as brain structure and cognitive function. Male mice are generally heavier than female mice, but upon HFD feeding we found that both sexes followed the same weight gain pattern. In our study, both sexes store fat in the inguinal fat depot, but male mice stored significantly more fat within the mesenteric fat depot than female mice resulting in more visceral fat in male mice. In line with this, male subjects tend to accumulate more fat in the visceral cavity, whereas female subjects accumulate more fat in the subcutaneous depot [13]. Visceral adipose tissue (VAT), including mesenteric fat, is associated with metabolic abnormalities, increased expression of inflammatory mediators and cardiovascular risk. In our study, male mice, fed either an AIN93G diet or HFD, had a higher inflammatory tone in the mesenteric fat depots than female mice which is consistent with results of others reporting that obese male mice develop more pronounced adipose tissue inflammation [45]. Furthermore, a time course analysis of adipose tissue expansion in adult C57BL/6 mice showed that visceral adipose tissue depots of male mice become inflamed once their maximal storage capacity has been reached [37]. It is unclear why the mass of perigonadal fat depot of young male Ldlr-/-.Leiden mice did not increase in response to HFD feeding. This depot is typically the first depot that expands and becomes inflamed in these mice, as well as in wild type C57BL/6 mice in general [28,46,47]. However, it is possible that in early life, molecular factors controlling adipogenesis are modulated by HFD in a different way on the epigenetic level than in adult mice [48]. Interestingly, adipose tissue inflammation was practically absent in female mice, despite a comparable (inguinal depot) or even greater (perigonadal depot) increase in mass, suggesting that either the storage capacity has not been reached in female mice, or that other mechanisms play a role in the onset of adipose tissue inflammation. It is also possible that the production of pro-inflammatory factors in VAT is greater in males than in females [16]. 

Sex differences regarding fat storage across the adipose tissue depots studied herein might at least partly be caused by differences in sex steroids. For instance, Kanaley and colleagues studied adipose tissue distribution in pre- and postmenopausal women and found that the postmenopausal drop in oestrogen was associated with a rapid increase in VAT accumulation and an increased risk of cardiovascular disease [49]. Moreover, in a knock-out mouse study, it was observed that the two oestrogen receptors (ER), ERα and ERβ, had distinct effects on adipose tissue distribution, which further supports a role of sex-steroids in fat storage across the various depots [50]. The potential modulating role of oestrogen in adipose tissue accumulation was suggested to be depot-specific because of the differential expression of ERα and ERβ within various WAT depots [50,51,52]. However, relatively little is known about the exact underlying mechanisms [16]; future research may concentrate on the role of sex steroids during HFD-induced obesity to gain more information on the modulating role of oestrogen in WAT depot expansions. In conclusion, our findings in young female and young male mice are in coherence with reported sex differences regarding fat distribution and inflammatory tone, the latter of which may be lower in females because of the anti-inflammatory properties of oestrogens [53]. Plausibly, the observed sex differences might be due to different expression of sex steroids, and differences in oestrogen receptor expression levels in adipose fat depots. 

NAFLD is known to develop during the course of diet-induced obesity [54,55]. In the current study, microvesicular steatosis increased in male mice only upon HFD feeding, whereas macrovesicular steatosis did not further increase. This differs clearly from adult Ldlr-/-.Leiden mice in which both forms of steatosis increased markedly upon HFD feeding [27,28]. The development of microvesicular steatosis is thought to be related to mitochondrial dysfunction, and malfunction of the mitochondrial β-oxidation of fatty acids [56,57,58]. It is possible that male mice are more susceptible to develop metabolic overload and metabolic stress in the liver, potentially because lipids are utilized to a lesser degree and accumulate in organs (either in adipose tissue for lipid storage or as ectopic fat). Elevated plasma triglyceride levels are a marker of NAFLD in humans and only a few diet-inducible NASH mouse models clearly recapitulate this aspect of the disease [25,59,60]. We found that in young male Ldlr-/-.Leiden mice triglyceride levels increased upon HFD, which differs from studies performed in wild type C57BL/6 mice, and in our study triglyceride levels and microvesicular steatosis were significantly correlated (r = 0.60, *p* < 0.001) [11]. In line with our findings, the complete absence of an effect of HFD feeding on plasma triglyceride levels in females was also reported by Hwang and colleagues in regular C57BL/6 mice [11].

Insulin resistance is associated with the development of NASH and with an increase in inflammatory cells within WAT and liver [27,61]. Consistent with this, we observed a strong increase of plasma insulin levels in young male mice on HFD, whereas young female mice showed surprisingly little response in insulin levels. Plasma insulin was found to be correlated with higher inflammatory tone in perigonadal (r = 0.64, *p* < 0.001) and mesenteric fat depots (r = 0.56, *p* < 0.001), and microvesicular steatosis (r = 0.56, *p* < 0.001) in male mice further supporting the use of male mice as a model for NAFLD/NASH [25,27,62]. Female mice appear to be less suited to mimic this facet of the etiology, which represents a great portion of the patients at risk of NAFLD/NASH. Notably, we found that macrovesicular steatosis and lobular inflammation were strongly present in young female mice independent of diet suggesting that young female mice store fat in a way that promotes the development of large vesicles. Clinical manifestations of NAFLD are typically characterized by predominantly macrovesicular steatosis in adulthood [63], but in children mixed forms of macro- and microvesicular steatosis are observed. In the case of macrovesicular steatosis, the hepatocyte contains a single, large vacuole, which fills up the hepatocyte and displaces the nucleus to the periphery of the cell [56,57]. Development of macrovesicular steatosis may be the result or combination of the following mechanisms: increased mobilization of fat from adipose tissue, increased de novo synthesis of fatty acids by the liver, increased fatty acid esterification into triglycerides and/or reduced utilization of triglycerides by the liver for energy metabolism, and impaired mitochondrial or peroxisomal β-oxidation [56,57]. In the present study macrovesicular steatosis was significantly associated with lobular inflammation in young male mice (r = 0.55, *p* < 0.01). Our finding is in line with similar findings in a study by Mulder and colleagues, who reported that HFD-induced macrovesicular steatosis is significantly associated with lobular inflammation in adult male mice [46]. In conclusion, our findings suggest that Ldlr-/-.Leiden male mice are more prone to respond to HFD feeding with hepatic steatosis development, based on the observed increase in microvesicular steatosis associated with hyperinsulinemia and hypertriglyceridemia, and the increase of the inflammatory tone in adipose tissue. Based on the observations made in this study, we believe that the general assumption that female and male mice should be representative for female and male patients, respectively, is incorrect, at least as a general statement. It may well be that a particular gender (e.g., female mouse) is more representative for male and female patients than the other gender (e.g., male mouse). For instance, in the field of atherosclerosis, it has been shown that female mice respond better to particular diets that promote hypercholesterolemia and atherogenesis and that male mice are not useful for these type of studies [64,65], and that observations made in female mice are highly predictive for male and female patients with cardiovascular disease [66,67]. These findings emphasize that the choice of a model should be made carefully, and it should be taken into account that human and mouse Y chromosomes differ clearly from each other (and from other mammals) in multiple ways as they carry different genes [68,69]. This knowledge should result in more critical considerations when it comes to the performance of preclinical studies that aim to investigate sex-specific differences in humans. 

Next to metabolically active tissues, sex-specific alterations in brain structure in the development of juvenile obesity were observed. We observed that white and gray matter integrity was reduced in juvenile male mice in various regions. Others reported sex differences in cerebral volume; for instance, female mice displayed larger anterior hippocampus volumes, while male mice had larger medial amygdala volumes [70]. Human studies have shown that young men (< 30 years) have larger amygdala and thalamus volumes compared to young women [71]. In addition, it has been reported that cerebral white matter increased linearly with age at a faster pace in women [71]. In our study in juvenile mice, we detected limited HFD-induced alterations in daytime activity and no significant alterations in cognitive function, which was indicated with the open field test and Morris water maze. However, others found HFD-induced circadian rhythm reprogramming and HFD-induced cognitive impairment in rodent models [7,8,72]. For instance, Boitard and colleagues found a decrease in hippocampal neurogenesis, memory impairment and an increase in hippocampal pro-inflammatory cytokines in juvenile obese mice [8]. Moreover, anxiety behavior was increased in juvenile obese mice compared to their lean controls [9]. In both studies, a HFD was induced immediately after weaning (3 weeks of age) [8,9], while in the current study a HFD was initiated at 6 weeks of age (3 weeks post-weaning). Therefore, the current model of initiating juvenile obesity differs in period of sensitivity; for instance, brain function is more susceptible to detrimental effects of HFD exposure earlier in life (3 weeks of age), or the brain is able to adapt to metabolic stressors from 6 weeks onwards. In our study, hippocampal CBF was reduced in HFD fed male mice as compared to AIN93G fed male mice. This finding is in agreement with previous research reporting a reduction in CBF in HFD-induced obese mice [28]. In humans, CBF is negatively associated with BMI in adults [73,74]. Furthermore, we found that the hippocampal white matter integrity was lower in HFD fed male mice. Previous research showed a correlation between hippocampal FA values and cognitive parameters in humans [75] and mice [76]. Thereby, a reduction in CBF and white matter integrity might be a precursor for the development of cognitive problems [77]. Nevertheless, in the current study, neuroinflammation, structural brain changes and CBF changes were still modest in young male mice, and might not yet have been large enough to provoke cognitive dysfunction. In more detail, in our study hepatic inflammation was positively associated with neuroinflammation (r = 0.67, *p* < 0.01) in male mice, whereas female mice lacked a significant correlation (r = −0.13). This correlation in male mice might suggest a link between the onset of hepatic inflammation and neuroinflammation. This is in line with findings by others, who report that hepatic inflammation may induce swelling of astrocytes and reactivation of microglia via an increased expression of inflammatory cytokines [78,79]. More specifically, inflammatory cytokines, such as interleukin-15, -6 or -1β, can promote brain infiltration, or transmit signals through interleukin receptors in endothelial cells in vascular and capillary structures of the brain [78]. This link between hepatic inflammation and neuroinflammation suggests that an increased expression of inflammatory cytokines by hepatic or WAT inflammation can induce neuroinflammation, alter the cerebrovasculature, or even affect brain function. Nevertheless, in our study the level of hepatic inflammation and neuroinflammation were modest in our juvenile male mice. We hypothesize that a prolonged HFD intervention or an earlier exposure of a HFD intervention (3 weeks of age) might induce a higher inflammatory tone in this mouse model. Future research should determine inflammatory cytokines levels in plasma and brain tissue, which eventually may reinforce the link between hepatic inflammation and neuroinflammation in context of obesity.

## 5. Conclusions

In conclusion, sex-specific differences observed in young mice fed a HFD include effects on (1) fat storage and distribution like increased mesenteric fat mass and adipose tissue inflammation specifically in male mice, (2) plasma levels of metabolic risk factors as in male mice hyperinsulinemia and hypertriglyceridemia, and (3) in the development of NAFLD including HFD-induced microvesicular steatosis only in male mice. Moreover, sex-specific differences in brain structure were observed as male mice displayed reduced CBF and white matter integrity within the hippocampus after HFD feeding. Surprisingly, female mice displayed macrovesicular steatosis and lobular inflammation in the liver when fed an AIN93G reference diet, and HFD feeding did not aggravate steatosis and inflammation, neither in liver nor adipose tissue. We conclude that young male Ldlr-/-.Leiden mice are more susceptible to the detrimental effects of a HFD than young female mice, resulting in male mice in increased VAT storage, adipose tissue inflammation, microvesicular steatosis in the liver, and reduced CBF and white matter integrity. Future research should focus on the role of sex steroids, oestrogen receptors and inflammatory cytokines in adipose tissue storage, the development of NAFLD, and plausible associations to brain function in the context of juvenile obesity. Eventually, this may help to develop sex-specific interventions and representative translational animal models. The current study emphasizes the importance of sex specific differences in the development of obesity, liver pathology and brain function, and advocates a careful choice of a model system. 

## Figures and Tables

**Figure 1 nutrients-11-01861-f001:**
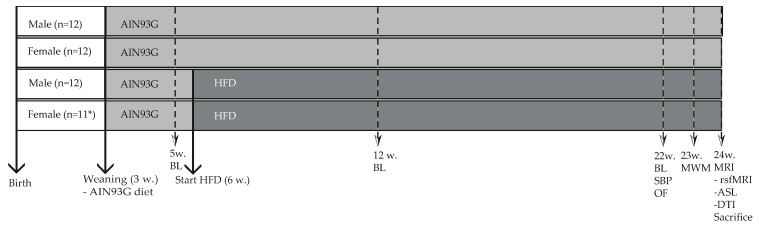
Study design: Food intake (at cage level) and body weight (individual) were weekly measured. Blood samples were taken after 5 h of fasting (8 a.m.–1 p.m.), and collected at three time points (5, 12 and 22 weeks (w)). Cognitive and MRI experiments were performed at 24 weeks of age. At the end of the experiment, all mice were anaesthetized and sacrificed by transcardial perfusion with 0.1M phosphate-buffered saline (PBS, 7.3 pH, room temperature). Thereafter, several organs (liver, adipose tissue and brain) were harvested and used for biochemical and immunohistochemical experiments. Gray: period fed a low-fat reference AIN93G diet. Deep gray: period of HFD feeding. BL = blood sample collection. MWM = Morris water maze test; SBP = systolic blood pressure measurements; OF = open field test; MRI = examined with magnetic resonance imaging for resting state fMRI (rsfMRI), arterial spin labeling (ASL) and diffusion tensor imaging (DTI). * 1 female mouse was sacrificed due to severe inflammation on the base of the tail.

**Figure 2 nutrients-11-01861-f002:**
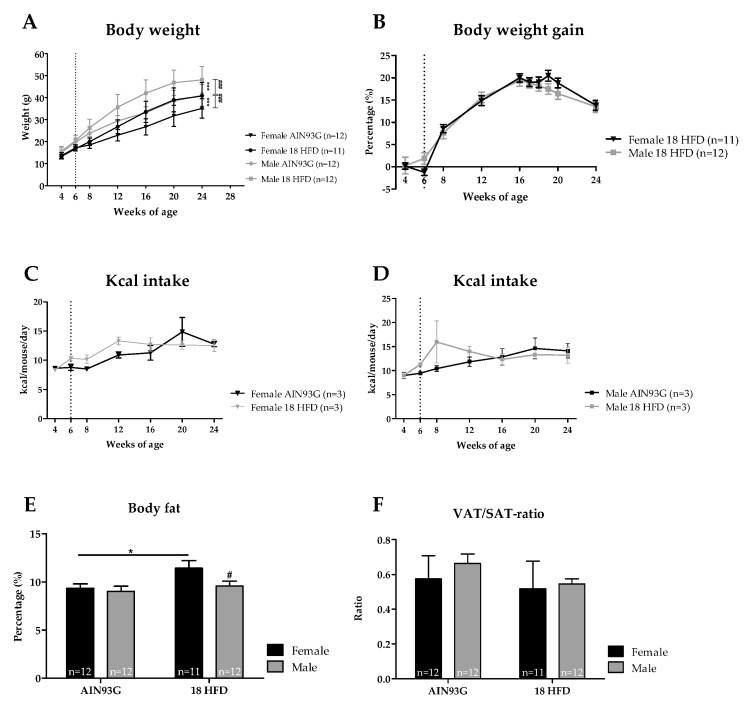
Analysis for body weight, food intake and body fat. (**A**) Absolute body weight in gram. (**B**) Body weight gain as percentage of initial weight of male and female mice fed a HFD after adjustment of the regular growth-related gain in the AIN93G control groups. Kcal intake is presented for female mice in (**C**), and for male mice in (**D**). (**E**) percentage of body fat calculated as the sum of perigonadal, mesenteric and inguinal fat pad weight divided by body weight. (**F**) VAT/SAT-ratio indicated by dividing the mesenteric weight by inguinal weight. The start of HFD feeding at 6 weeks of age is indicated as a vertical, dotted line. * *p* ≤ 0.05 and *** *p* ≤ 0.001 significant diet effect (HFD versus AIN93G), ^#^
*p* ≤ 0.050, ^###^
*p* ≤ 0.001 significant sex-specific effect.

**Figure 3 nutrients-11-01861-f003:**
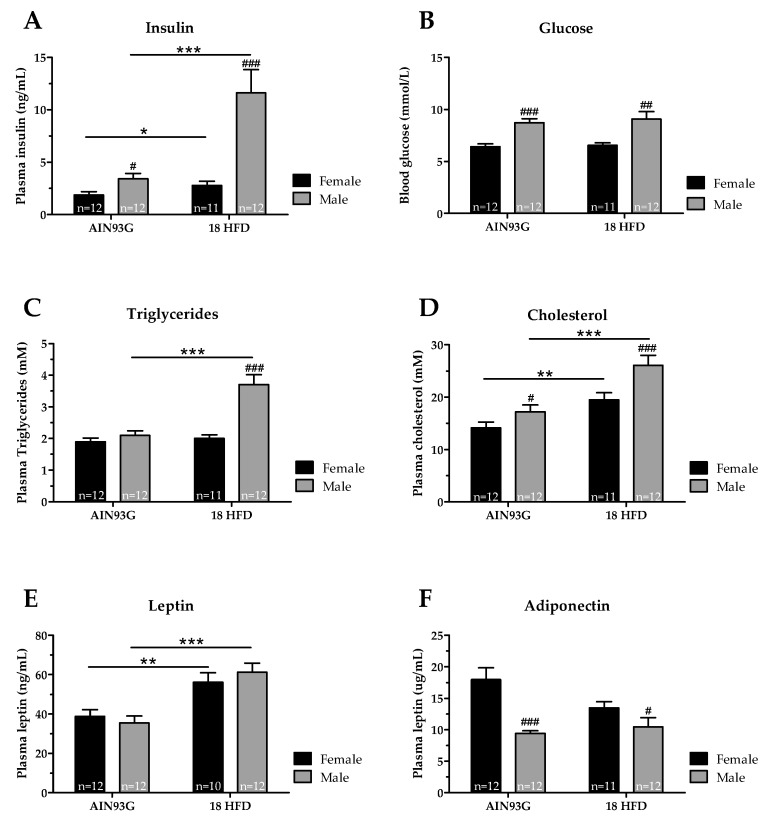
Circulating levels of metabolic risk markers in Ldlr-/-.Leiden mice after 18 weeks of AIN93G diet or HFD feeding. Five-hour fasted levels of (**A**) insulin, (**B**) glucose, (**C**) triglycerides, (**D**) cholesterol, (**E**) leptin and (**F**) adiponectin. * *p* ≤ 0.050, ** *p* ≤ 0.01 and *** *p* ≤ 0.001 significant dietary effect. ^#^
*p* ≤ 0.050, ^##^
*p* ≤ 0.010 and ^###^
*p* ≤ 0.001 significant effect in sex.

**Figure 4 nutrients-11-01861-f004:**
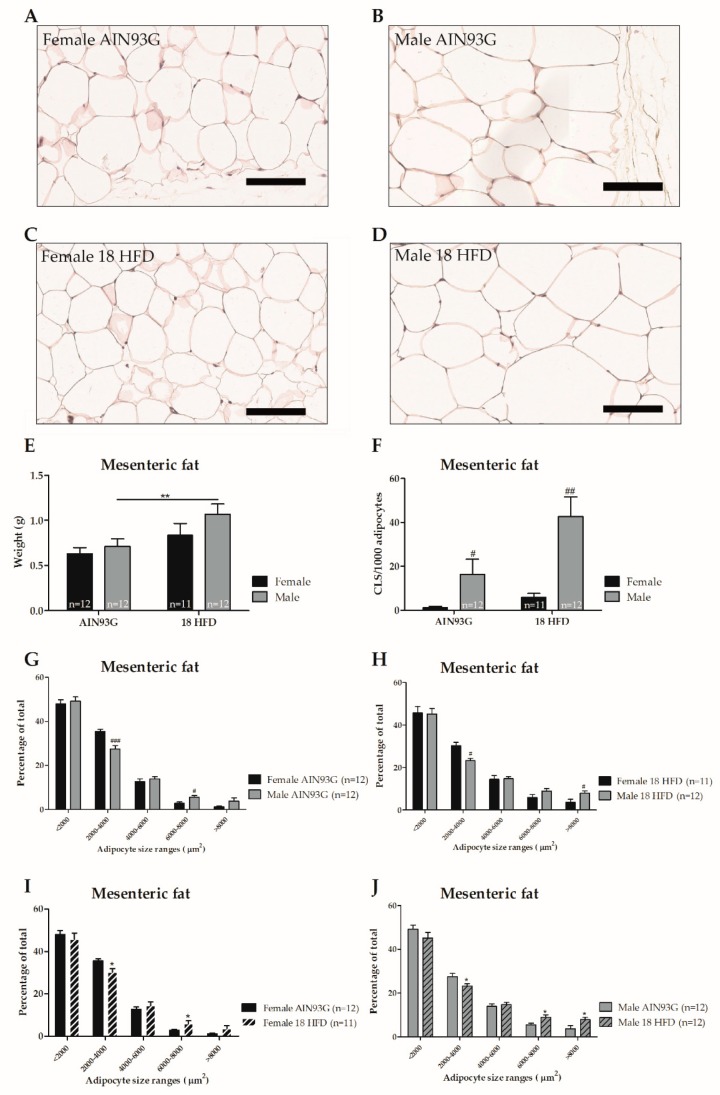
Adipose tissue analysis in Ldlr-/-.Leiden mice after 18 weeks of AIN93G diet or HFD feeding. Representative photomicrographs of haematoxylin phloxine and saffron (HPS) stained mesenteric adipose tissue per experimental group (**A**–**D**). Scale bar is 100 µm. Analyses of weight (**E**) and level of inflammation as indicated by crown-like structures (CLS) per 1000 adipocytes (**F**) in the mesenteric fat depot. Distribution of adipocyte sizes for either AIN93G diet (**G**), HFD diet (**H**) or sex: female (**I**) and male (**J**). Significant dietary effects * *p* ≤ 0.050 and ** *p* ≤ 0.010. Significant effects in sex ^#^
*p* ≤ 0.050 and ^###^
*p* ≤ 0.001.

**Figure 5 nutrients-11-01861-f005:**
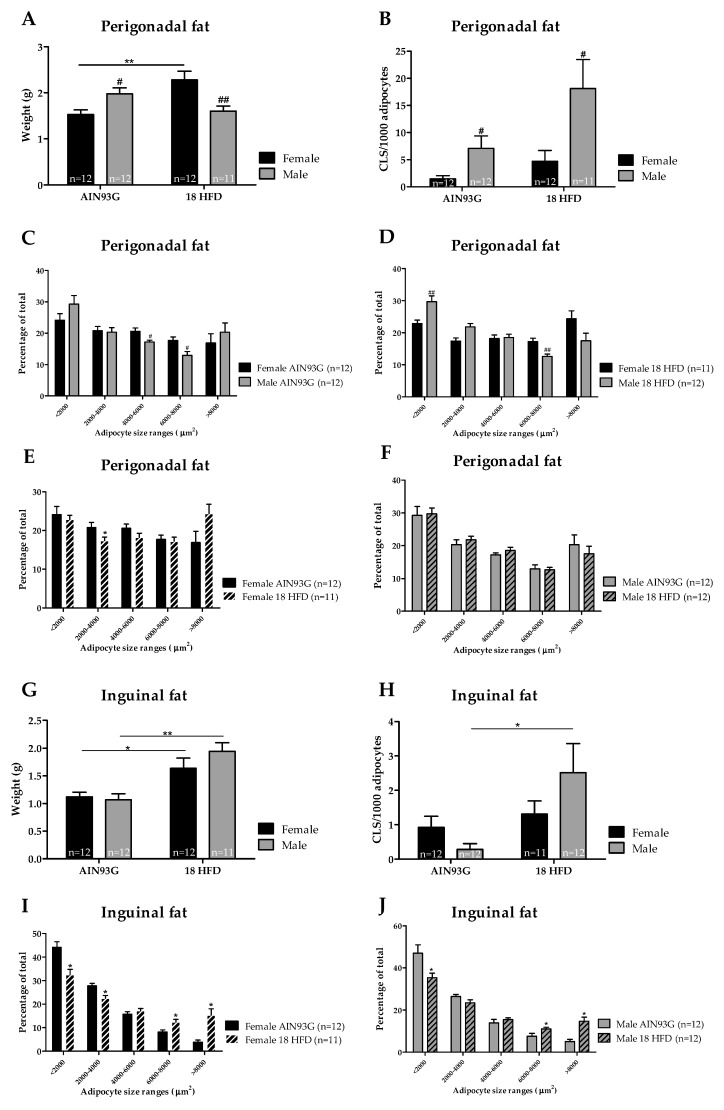
Analysis of perigonadal and inguinal fat depots in Ldlr-/-.Leiden mice after 18 weeks of AIN93G or HFD feeding. Perigonadal fat depot (**A**) weight, and (**B**) adipose tissue inflammation as indicated by crown-like structures (CLS) per 1000 adipocyte. Distribution of adipocyte sizes per diet (**C**) AIN93G, (**D**) HFD, or for (**E**) female mice and (**F**) male mice. The inguinal fat depot in (**G**) weight, and (**H**) adipose tissue inflammation as indicated by crown-like structures (CLS) per 1000 adipocytes. The distribution of adipocyte sizes for (**I**) female mice and (**J**) male mice. * *p* ≤ 0.050, ** *p* ≤0.010, and *** *p* ≤ 0.001 significant dietary effect (HFD versus AIN93G). ^#^
*p* ≤ 0.050 and ^##^
*p* ≤ 0.001 significant sex-specific effect.

**Figure 6 nutrients-11-01861-f006:**
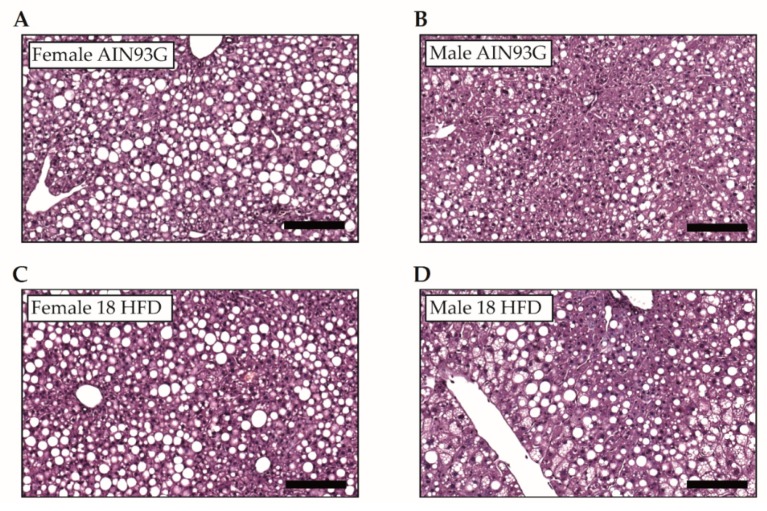
Analyses of histological hallmarks of non-alcoholic fatty liver disease after 18 weeks of AIN93G diet or HFD feeding in young Ldlr-/-.Leiden male and female mice. Representative images of haematoxylin and eosin-stained liver tissue per experimental group (**A**–**D**). Scale bar is 100 µm. (**E**) Liver weight, (**F**) microvesicular steatosis, (**G**) macrovesicular steatosis, (**H**) hepatocellular hypertrophy, (**I**) hepatic inflammation. * *p* ≤ 0.050 and *** *p* ≤ 0.001 significant dietary effect. ^#^
*p* ≤ 0.050, ^##^
*p* ≤ 0.010 and ^###^
*p* ≤ 0.001 significant effect in sex.

**Figure 7 nutrients-11-01861-f007:**
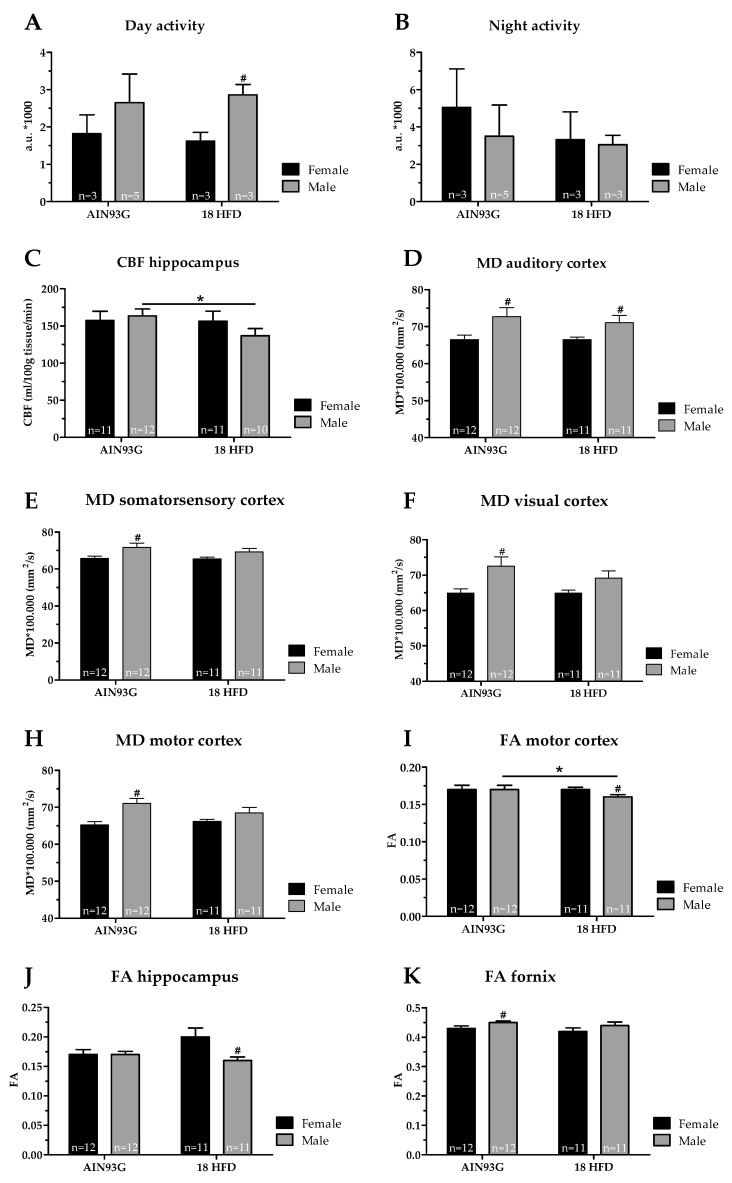
Activity and brain structure in Ldlr-/-.Leiden mice after 18 weeks of AIN93G diet or HFD. Day activity (**A**) and night activity (**B**) were indicated by activation of 12 cage-sensors and expressed in arbitrary units (a.u.) per mouse at cage level. (**C**) Cerebral blood flow (CBF) within the hippocampus. Mean diffusivity (MD) levels within the (**D**) auditory cortex, (**E**) somatosensory cortex, (**F**) visual cortex, and (**H**) motor cortex. Higher MD levels are associated with a decreased gray matter integrity. Fractional anisotropy (FA) levels within the (**I**) motor cortex, (**J**) hippocampus and (**K**) fornix. Lower FA levels are associated with a decreased white matter integrity. * *p* ≤ 0.050 significant dietary effect (HFD versus AIN93G), ^#^
*p* ≤ 0.050 significant sex-specific. Due to imaging artifacts, one female mice on AIN93G diet and two male mice on HFD were excluded in the CBF analysis, and one male mice fed HFD was excluded in the DTI analysis.

**Table 1 nutrients-11-01861-t001:** Dietary compositions. Presented are the proportions of certain substances within the diet mixture in percentages based on mass (gm%) and kilo calories (kcal %). AIN93G (D10012G) and high fat diet (HFD, D12451) were produced by Research Diets, Inc., New Brunswick, USA.

	AIN93G		HFD	
	gm%	kcal%	gm%	kcal%
Protein	20	20.3	24	20
Carbohydrate	64	63.9	41	35
Fat	7	15.8	24	45
Total		100		100
Kcal/gm	3.9		4.73	
Ingredient				
Casein, 30 Mesh	200	800	200	800
L-Cystine	3	12	3	12
Corn starch	397	1590	72.8	291
Maltodextrin	132	528	100	400
Sucrose	100	400	172.8	691
Cellulose	50	0	50	0
Soybean oil	70	630	25	225
Lard			177.5	1598
t-Butylhydroquinone	0.0014	0		
Di-calcium phosphate			13	0
Calcium carbonate			5.5	0
Potassium citrate			16.5	0
Choline bitartrate	2.5	0	2.0	0
Mineral mix S10022G	35	0		
Mineral mix S10026			10	0
Vitamin mix V10037	10	40		
Vitamin mix V10001			10	40
FD&C red dye #40			0.05	0
**Total**	1000	4000	858.15	4057

**Table 2 nutrients-11-01861-t002:** Imaging parameters.

	Anatomical T2 * w	ASL	DTI	rsfMRI
Imaging method	GE	FAIR-ASL	4-shot spin-echo PI	Spin-echo EPI
Echo time (ms)	7.35	10.8	21.0	10.0
Repetition time (s)	0.86	12.0	21.0	1.85
Image matrix (pixel × pixel)	512 × 512	128 × 96	128 × 128	96 × 96
Field-of-view (mm)	40 × 40	25 × 25	20 × 20	25 × 25
Spatial resolution (µm/pixel)	78 × 78 × 500	195 × 260 × 1000	156 × 156 × 500	260 × 260 × 500
Number of slices	20 × 3	16	20	20
Total acquisition time (min)	8	12	35	11

Anatomical reference scans were T2 star weighed (T2*w).

**Table 3 nutrients-11-01861-t003:** Overview of *p*- and *F*-values of significant effects between male and female mice fed either a AIN93G diet or 18 weeks of high fat diet (HFD).

Parameters	AIN93G	HFD
	Female Versus Male	Female Versus Male
Body weight	↓ *p < 0.001, F(1,22) = 25.98*	↓ *p < 0.001, F(1,22) = 17.22*
Body weight gain	=	=
Food intake	=	=
Percentage of body fat	=	↑ *p < 0.050, F(1,21) = 4.19*
VAT/SAT-ratio	=	=
**Fasting plasma levels**		
Glucose	↓ *p < 0.001, F(1,22) = 24.43*	↓ *p = 0.010, F(1,20) = 7.98*
Insulin	↓ *p = 0.015, F(1,22) = 6.97*	↓ *p < 0.001, F(1,20) = 25.50*
Cholesterol	=	↓ *p = 0.018, F(1,20) = 6.65*
Triglycerides	=	↓ *p < 0.001, F(1,20) = 30.20*
Adiponectin	↑ *p < 0.001, F(1, 22) = 33.51*	↑ *p = 0.020, F(1,20) = 6.34*
**Adipose tissue**		
Mesenteric weight	=	=
Mesenteric adipocyte size		
Averaged	=	=
Distribution, <2000 µm^2^	=	=
2000–4000 µm^2^	↑ *p < 0.001, F(1,22) = 19.90*	↑ *p = 0.001, F(1,21) = 13.98*
4000–6000 µm^2^	=	=
6000–8000 µm^2^	↓ *p < 0.010, F(1, 22) = 9.50*	=
>8000 µm^2^	=	*↓p = 0.036, F(1,21) = 5.01*
Mesenteric CLS	↓ *p = 0.012, F(1,22) = 7.48*	↓ *p < 0.001, F(1,21) = 17.44*
Perigonadal weight	↓ *p = 0.020, F(1,22) = 5.84*	↑ *p = 0.007, F(1,21) = 9.08*
Perigonadal adipocyte size		
Averaged in µm^2^	=	↑ *p = 0.026, F(1,21) = 5.70*
Distribution, <2000 µm^2^	=	↓ *p = 0.001, F(1,21) = 13.98*
2000–4000 µm^2^	=	=
4000–6000 µm^2^	↑ *p < 0.013, F(1, 22) = 7.21*	=
6000–8000 µm^2^	↑ *p < 0.005, F(1, 22) = 9.70*	↑ *p = 0.003, F(1,21) = 10.85*
>8000 µm^2^	=	=
Perigonadal CLS	=	↓ *p = 0.050, F(1,21) = 4.34*
Inguinal weight	↓ *p = 0.002, F(1,22) = 11.86*	=
Inguinal adipocyte size		
Averaged in µm^2^	=	=
Distribution in µm^2^	=	=
Inguinal CLS	↓ *p = 0.027, F(1,22) = 5.63*	↓ *p = 0.034, F(1,21) = 5.13*
>8000 µm^2^		
**Liver**		
Liver weight	=	↑ *p = 0.022, F(1,22) = 6.05*
Microvesicular steatosis	=	↓ *p < 0.001, F(1,18) = 31.61*
Macrovesicular steatosis	↑ *p < 0.001, F(1,21) = 27.31*	↑ *p = 0.002, F(1,18) = 13.13*
Inflammation	↑ *p < 0.001, F(1,21) = 48.59*	↑ *p < 0.001, F(1,18) = 38.65*
Hypertrophy	=	↓ *p = 0.043, F(1,18) = 4.74*
**Cognition**		
Open field	=	=
Morris water maze	=	=
DVC	=	↓ *p = 0.027, F(1,4) = 11.72*
SBP	=	=
**MRI**		
Blood vessels (GLUT-1)	=	=
Cerebral blood flow	=	=
Neurogenesis (DCX)	=	=
Neuroinflammation (IBA-1)	=	=
Functional connectivity	=	=
DTI, mean diffusivity		
Auditory cortex	↓ *p = 0.030, F(1,22) = 5.37*	↓ *p = 0.041, F(1,20) = 4.77*
Somatosensory cortex	↓ *p = 0.038, F(1,22) = 4.86*	=
Visual cortex	↓ *p = 0.013, F(1,22) = 7.22*	=
Motor cortex	↓ *p = 0.002, F(1,22) = 12.60*	=
DTI, fractional anisotropy		
Hippocampus	=	↑ *p = 0.033, F(1,20) = 5.24*
Motor cortex	=	↑ *p = 0.013, F(1,20) = 7.42*

↑: Female mice have a significant higher level than male mice. ↓: Female mice have a significant lower level than male mice. =: No significant differences between female or male mice.

**Table 4 nutrients-11-01861-t004:** Overview of *p*- and *F*-values of significant effects between mice fed AIN93G diet or 18 weeks of high fat diet (HFD) for either male or female mice.

Parameters	Female	Male
	HFD versus AIN93G	HFD versus AIN93G
Body weight	↑ *p = 0.004, F(1,21) = 10.31*	↑ *p = 0.002, F(1,22) = 12.11*
Food intake	=	=
Percentage of body fat	↑ *p = 0.026, F(1,21) = 5.69*	=
VAT/SAT-ratio	=	=
**Fasting plasma levels**		
Glucose	=	=
Insulin	↑ *p = 0.035, F(1,20) = 5.11*	↑ *p < 0.001, F(1,22) = 19.72*
Cholesterol	↑ *p = 0.007, F(1,20) = 8.88*	↑ *p = 0.001, F(1,22) = 14.83*
Triglycerides	=	↑ *p < 0.001, F(1,22) = 25.47*
Adiponectin	=	=
Leptin	↑ *p = 0.006, F(1,20) = 9.59*	↑ *p < 0.001, F(1,22) = 19.75*
**Adipose tissue**		
Mesenteric weight	=	↑ *p = 0.045, F(1,22) = 4.50*
Mesenteric adipocyte size		
Averaged	=	=
Distribution, < 2000 µm^2^	=	=
2000–4000 µm^2^	↓ *p = 0.008, F(1,21) = 8.59*	↓ *p = 0.036, F(1,22) = 4.98*
4000–6000 µm^2^	=	=
6000–8000 µm^2^	↑ *p = 0.036, F(1,21) = 5.00*	↑ *p = 0.016, F(1,22) = 6.88*
>8000 µm^2^	=	=
Mesenteric CLS	=	↑ *p = 0.016, F(1,22) = 6.86*
Perigonadal weight	↑ *p = 0.003, F(1,21) = 11.78*	=
Perigonadal adipocyte size		
Averaged	=	=
Distribution, <2000 µm^2^	=	=
2000–4000 µm^2^	↓ *p = 0.047, F(1,21) = 84.43*	=
4000–6000 µm^2^	=	=
6000–8000 µm^2^	=	=
>8000 µm^2^	=	=
Perigonadal CLS	=	=
Inguinal weight	↑ *p = 0.030, F(1,21) = 5.17*	↑ *p = 0.002, F(1,22) = 11.86*
**Inguinal adipocyte size**		
Averaged	↑ *p < 0.001, F(1,21) = 22.82*	↑ *p = 0.002, F(1,22) = 12.80*
Distribution, < 2000 µm^2^	↓ *p = 0.002, F(1,21) = 12.31*	↓ *p = 0.017, F(1,22) = 6.70*
2000–4000 µm^2^	↓ *p = 0.003, F(1,21) = 10.86*	
4000–6000 µm^2^	↑ *p = 0.008, F(1,21) = 8.67*	=
6000–8000 µm^2^	↑ *p < 0.000, F(1,21) = 19.43*	↑ *p = 0.032, F(1,22) = 5.22*
>8000 µm^2^	=	↑ *p = 0.002, F(1,22) = 12.80*
Inguinal CLS	=	↑ *p = 0.016, F(1,22) = 6.82*
**Liver**		
Liver weight	=	↑ *p = 0.022, F(1,22) = 6.05*
Microvesicular steatosis	=	↑ *p < 0.001, F(1,18) = 23.47*
Macrovesicular steatosis	=	=
Inflammation	=	↑ *p < 0.001, F(1,21) = 48.59*
Hypertrophy	=	↑ *p = 0.013, F(1,21) = 7.32*
**Cognition**		
Open field	=	=
Morris water maze	=	=
Digital ventilated cages	=	=
Systolic blood pressure	=	=
**MRI**		
Blood vessels (GLUT-1)	=	=
Cerebral blood flow	=	↓ *p = 0.010, F(1,17) = 8.44*
Neurogenesis (DCX)	=	=
Neuroinflammation (IBA-1)	=	=
Functional connectivity	=	=
DTI, mean diffusivity	=	=
DTI, fractional anisotropy		
Motor cortex	=	↓ *p = 0.045, F(1,21) = 4.55*

↑: HFD fed mice, either female or male, have a significant higher level compared to AIN93G fed mice. ↓: HFD fed mice, either female or male, have a significant lower level compared to AIN93G fed mice. =: No significant differences between diets.

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
