# Peer review of "Sex-Specific Differences in Fat Storage, Development of Non-Alcoholic Fatty Liver Disease and Brain Structure in Juvenile HFD-Induced Obese Ldlr-/-.Leiden Mice"

_nutrients, 2019, doi:10.3390/nu11081861_

Round 1

Reviewer 1 Report

The study by Jacobs et al. describes the sex differences that occur resulting from a model of juvenile high fight diet induced obesity.  The authors investigate fat distribution and storage, measures of metabolism, liver pathology and brain structure.  The authors conclude that male mice are more negatively influenced by high fat diet than female mice.  The strengths of this study are its breadth in investigating different aspects of pathology during this disorder.  Unfortunately, this led to some aspects of the study that I feel could be improved.  These are outlined below:

1)      Many of the analyses and data presented throughout are shown in bar graph form when morphometry and histological analyses are performed.  These include adipocyte size, H&E stains in liver and others.  Representative images that the pathologist was using for analysis should be provided for these analyses.  Without seeing these images, I cannot make further suggestions on other areas that may need investigation.

2)      In the abstract and methods, behavioral testing is described including open field and Morris water maze but the data is not shown or discussed in the methods.  This data should be included to link the changes in brain structure to functional changes in behavior.

3)      Near the end of the discussion the authors mention a possible connection in male mice between hepatic inflammation and neuroinflammation that could be similar to what is seen during hepatic encephalopathy (lines 541-547).  However, in the results the authors mention there was no change in neuroinflammation caused by diet or sex as assessed by IBA1 (lines 387-388).  This seems contradictory.  The authors need to include other measures of neuroinflammation such as cytokine data to better clarify and support these statements.  In addition, the authors should include the IBA1 and GLUT-1 analyses in the paper or as supplemental data.

4)      When discussing hepatic injury and the development of NASH, the authors should better characterize hepatic function and inflammation.  The authors need to perform serum chemistry analyses at minimum (ALT, AST, bilirubin) but assessing serum albumin and cytokines would strengthen the conclusions regarding the liver.

Reviewer 2 Report

The authors investigated the sex-specific effects of HFD-induced obese in juvenile mice adipose tissues, liver and brain. Male mice are more prone to HFD-induced detrimental effects regarding fat distribution, inflammatory responses and structural brain changes.

General comments:

The manuscript is well structured and data well presented, and the authors did a thorough work on analyzing results and discussing in the context of previously published studies. Therefore, here I have only minor comment:

The figure resolution is not good enough, and some texts in the figures are difficult to read, e.g. the numbers of adipocyte size range in Fig. 4 and Fig. 5. I would suggest the authors save/upload their figures in the format of vector image.

Specific comments:

The supplementary table 1 is missing.

Fig. 2A. The icons for AIN93G and HFD (within same sex) look identical to me, as it is impossible to distinguish these two diets in the figure.

Line 404 and 406, ‘grey matter’ should be ‘white matter’?

Round 2

Reviewer 1 Report

The authors have addressed most of my criticisms on the resubmission of their study.  In regards to the analyses they could not perform (serum chemistry measures), they gave rationale and support to why this would not influence overall outcomes of this study.  Based upon the changes to the manuscript and the responses to the reviewers, I think the manuscript is sound and robust though the authors may want to look into the following minor change below:

With the new added histological and immunological images (figures 4A-D and figures 6A-D) , the authors should label the top of the figures with the groups (sex and diet) to make it clearer for the readers.
